# Older People’s Experiences of Living with, Responding to and Managing Sensory Loss

**DOI:** 10.3390/healthcare9030329

**Published:** 2021-03-15

**Authors:** I Ching Ho, Lynn Chenoweth, Anna Williams

**Affiliations:** 1Faculty of Medicine, University of New South Wales, Sydney, NSW 2052, Australia; 2Centre of Healthy Brain Ageing (CHeBA), Faculty of Medicine, Sydney, NSW 2052, Australia; l.chenoweth@unsw.edu.au; 3School of Nursing, University of Notre Dame Australia, Darlinghurst, NSW 2010, Australia; anna.williams@nd.edu.au

**Keywords:** older age, senses, sensory change, qualitative analysis

## Abstract

(1) Background: Ageing is associated with a decline in sensory function (sight, hearing, taste, touch and smell), which play an important role in the maintenance of an older person’s health, independence and well-being. (2) Methods: This qualitative study obtained data through face-to-face semi-structured interviews with a convenience sample of thirteen community-dwelling adults 65 years and older. Themes were derived inductively, guided by semi-structured interviews. (3) Results: Twelve participants had two or more sensory impairments, mainly concurrent hearing and vision, which became apparent when a situation/individual alerted them to change/s occurring. They were less aware of impaired smell, taste and touch. Sensory changes impacted on important life functions, prompting many participants to take measured risks in maintaining their independence. Half (seven) of the participants lacked motivation to manage sensory function through goal-directed behaviour, taking remedial actions only when this was relevant to lifestyle preferences. (4) Conclusions: Internal and/or external triggers of sensory changes did not generally motivate remedial action. Health professionals can help to improve older people’s attention to sensory impairment by routinely discussing sensory function with them, screening for sensory changes and facilitating early intervention and support.

## 1. Introduction

While declining sensory function in older people has been objectively measured and widely reported [1], what is less understood is how sensory changes impact on the person’s health, function, independence and well-being [2]. Objective measures of sensory impairments are helpful in identifying remaining sensory function; however, they cannot identify the burden of sensory impairment on older adult’s daily life and function. Two individuals with the same audiometric results may have different challenges and demands in their daily lives depending on personal, environmental and social factors [3]. Qualitative investigation of how sensory change is experienced and managed by the older person can provide insight into the lived experience, perceptions around adaptation and the facilitators and barriers to proactive management of sensory change. 

To our knowledge, only six qualitative studies have reported older people’s perceptions and personal experiences of living with, adapting to and managing sensory changes [4,5,6,7,8,9]. In these six studies, an official diagnosis of profound sensory impairment and an objective score of sensory function were identified for study participants who were attending community-based support services (e.g., low vision support groups). As these study participants were members/clients of targeted sensory support services, study findings may not reflect the experiences and management of sensory change/s for older people who do not access such services. Moreover, five of these studies reported findings on the impact and/or management of a decline in only one of the primary senses (either vision or hearing loss). Only one study [4] identified the impact of concurrent sensory changes on the older person’s lived experience with these losses. 

Intact sensory capacity is essential in shaping perceptions of information and stimuli, and for processing information in a meaningful way [10]. During the ageing process, these capacities are likely to diminish with the depreciation of the sensory functions of sight, hearing, taste, touch and smell [11]. Changes in sensory function are associated with poorer functional status [12], depression [13], increased risk of social isolation [14] and increased mortality rates [15]. A decline in one or more of the senses in older age, therefore, makes it harder for the person to adjust to changes in their bio-psychosocial environment [10].

As a decline in sensory function is common in older persons, it can often be normalised [16]. Sensory changes typically develop gradually, causing the person to adapt to the changes or ignore the losses occurring [17]. The perceived degree of sensory impairment can be influenced by the individual’s interactions with their physical and social environment [17]. A lack of recognition of declining senses (particularly sight and hearing) may have deleterious consequences for communication, socialisation and ability to live independently in older age [18]. Additionally, the denial of declining sensory function in older age may be inadvertently reinforced by family and friends, as they adapt to the changes occurring in the older person’s biopsychological functioning [2]. When the older person’s social and emotional connections remain strong despite declining sensory function, their self-perception and acceptance of sensory change/s occurring may be minimised. Compared to clinical measurement of sensory change/s, the older person’s perception of the sensory change/s occurring may be underrated [16,19]. 

The underrated perception of sensory changes is an important consideration, as while many older adults experience a change in all senses over time, some experience changes in only a few. In a population-based study of 2968 community-dwelling American adults aged 57 to 85, 38 percent had two sensory impairments and 28 percent had three or more sensory impairments [1]. The synergistic impact of concurrent impairments is due to the individual’s diminished biological resources and capacities, interfering with their ability to compensate for one sensory domain with another [20]. Concurrent sensory impairments may limit the usefulness of modern technologies which seek to overcome one sensory impairment by utilising other senses to retain functionality [21]. For example, compensation of hearing loss through presenting audio information in a visual format (e.g., television closed captioning) will be ineffective. 

New developments in self-service technology such as enhanced auditory applications are, however, becoming increasingly accessible to older adults [22]. There is now a wide range of age-friendly technology which can assist older people to self-manage chronic sensory impairment [23]. The technology roadmap for the Australian aged care sector [24] identified that a majority of older people are willing to embrace such technology as part of their overall healthcare plan, so long as the following conditions are met: (1) affordability; (2) control over the technology; (3) protection of privacy; (4) choice of use; (5) safety; (6) relevance to needs; (7) usability and reliability; and (8) integration into the home and everyday living. In general, an older person’s attitudes towards adopting new technology depend on four attributes: fit to daily life, trust in the delivered benefits, user-driven design and expected privacy [25]. Intent to use a particular self-service technology, however, is related to knowledge of the particular technology and the expected benefits directly associated with its use [26]. In turn, expected benefits and usefulness of a particular technology are influenced by control beliefs (i.e., context, flexibility and economic considerations), self-assessed capability (i.e., self-efficacy) and resource availability (i.e., facilitating conditions) [26]. In circumstances where the older person experiences concurrent sensory impairment, e.g., hearing and vision, which may have a compounding effect on their functional independence [27], self-service technologies may be beneficial.

Some older persons proactively modify their behaviour to maintain their independence and self-determination in the presence of sensory changes. However, behaviour modification may be adaptive or maladaptive [10]. Adaptive behaviour includes having regular health checks, maintaining personal relationships and making adjustments in the living environment [10]. According to the Goal-Directed Behaviour Model [28], adaptation to changes in health status require an individual to recognise that changes are occurring (such as a decline in one or more of the senses), acknowledge and accept the changes, and have a desire to maintain or regain health function. Thus, the individual plans an approach and takes action to address health issues and continues to seek solutions by evaluating the outcomes of actions taken [5]. When the older person lacks motivation to plan and carry out actions to address sensory change/s, maladaptive behaviour may occur, including self-isolation through avoidance of social and other situations which require the person to make use of failing senses, e.g., shopping. Such maladaptive lifestyle changes may provide short-term solutions to avoiding situations which require acknowledgement and taking action to address sensory issues, but they may not be effective in the long-term [20]. For example, turning up the volume of the television or moving closer to the television to compensate for the reduced auditory input delays the requirement to seek more effective solutions to hearing loss. 

The Goal-Directed Behaviour Model [28] suggests that if the environmental demands are either above or below the individuals’ competence level to take action on the changes occurring, they will more readily succumb to sensory changes, and will not take remedial action, engage in preventive measures or seek therapies and treatment to prevent further sensory loss. For example, having limited knowledge of self-management options, combined with reduced access to relevant healthcare support and the high cost of remedial aids, may result in inaction on sensory deterioration [7]. Moreover, features of the older person’s physical and social environment may also compromise the effectiveness of technological aids, causing the person to lose motivation in seeking remediation [29].

To learn more about the older person’s lived experience and adaptations to sensory impairment, four research questions were posed:To what extent is sensory change apparent to older people?Which of the senses is the most important to older people?What is the older person’s experience of living with sensory change?How do older people respond to and manage sensory change?

## 2. Materials and Methods

### 2.1. Design

A qualitative descriptive study using face to face semi-structured interviews. 

### 2.2. Setting

Five community-based adult day centres in metropolitan Sydney (Australia) provided access to older citizens who attended the centres’ exercise and social programs. 

### 2.3. Ethics

The study protocol was approved by the University of New South Wales (UNSW, Sydney, NSW, Australia) Human Research Ethics Committee (HC180855), the South Eastern Sydney Local Health District Human Research Ethics Committee (2019/ETH00213) and the executive teams of the five participating adult day centres. 

### 2.4. Participants and Recruitment

The convenience sample of adult day centre attendees was required to be 65 years and older, live independently in the community, be able to understand written and spoken English, be living with at least one sensory change (sight, hearing, taste, touch and/or smell) and be willing to consent to study participation. Participants were recruited using an arms-length approach in compliance with the National Statement on Ethical Conduct in Human Research (3.1.18 and 3.1.21) [30]. Details of 11 organisations offering community-based programs for older citizens operating in metropolitan Sydney were identified through the Australian Government’s “MyAgedCare” website and the Local Information Network for Community Services (LINCS) Directory Database. Program Directors were initially contacted through email to be informed about the study and to seek their interest. Five adult day centre directors agreed to display the study flyers at 13 of their centres which provided community programs for older citizens. Seven other program directors did not reply to the email, or to follow-up telephone calls to centre management. The study flyer provided an explanation of the study, including the study aims, interview procedure, researcher names and contact details. Interested day centre clients were asked to contact the lead researcher (IH) directly and nominate themselves to join the study. Six clients contacted the lead researcher directly through email or telephone, and seven clients associated with one centre notified the centre manager of their interest, who then passed the information on to the research team. 

The approved participant information statement and consent form (PISCF) was provided to interested clients by the centre manager via email or mail several days before the proposed interview. At the commencement of the planned interview time with older persons who expressed interest, the researchers provided further details of the study aims, procedures and intended outcomes prior to obtaining their consent, including permission to audio-record the interviews. Participants were made aware that they were free to withdraw from the study at any time with no penalties. Time was given for participants to ask and clarify any questions regarding the study before being asked to sign the PISCF. Participants were given a signed copy of the PISCF. Electronic copies of the PISCF were stored in a password-protected computer system on the secure university server.

### 2.5. Research Team

The research team comprised one female fourth year medical student at an Australian university (IH), and two experienced female academic and clinical researchers (LC and AW), who have considerable experiences in conducting both qualitative and quantitative studies, including clinical trials. Their research has focused on ageing transitions, aged and primary healthcare systems, chronic illness prevention and management and older person care models, nursing education and policy development. The research team had no previous relationship with the study participants. 

### 2.6. Participant Interviews

Semi-structured, face-to-face interviews were undertaken with participants regarding their perceptions of living with and responding to sensory changes. An interview guide (see Appendix A) was constructed by the researchers with reference to the literature on age-related sensory change experiences and responses. The semi-structured, open-ended questions represented a schematic presentation of topic questions relevant to the research questions, which helped to ensure that the same areas of information were collected from each participant, while allowing a degree of freedom and adaptability with information sharing. Interview questions included: background demographic data; sensory change history; awareness and identification of sensory change/s; experiences of living with sensory change/s; goal setting and responses to sensory function; management of the senses and associated factors; and any additional information the participants wanted to share. The draft interview questions were piloted with one participant, who provided verbal and written advice on the focus, structure, and wording of the interview questions, informing finalisation of the interview questions, e.g., clarification of concepts with examples, and interview process, e.g., interview setting, question order and timeframe.

### 2.7. Interview Procedures

All interviews were audio-recorded with participant agreement using two audio-recording devices (one for back-up, in case either device failed to record). For training purposes, eight interviews were conducted by the first author (IH), while the supervising researchers (LC or AW) also hand-recorded participant responses and assisted with probing interview responses. The remaining five interviews were conducted and audio-recorded by IH independently. Twelve of the 13 interviews were conducted in private closed rooms at the community day centres on dates and times that were decided by the participants. Upon participant request, one interview was conducted by IH over the phone in a private university office and audio-recorded. The interviews ranged in length from 30 to 90 min, depending on participant fatigue level and participant request to cease, or continue with, the interview. Data saturation was achieved by the ninth interview and was confirmed by four further interviews.

### 2.8. Data Entry and Storage

Interviews were transcribed verbatim by IH, who then checked transcript accuracy by listening to and cross checking the transcriptions with the audio-recordings. Inadvertent recording of the community centre, day care program staff or participant names were erased from the audio-recordings and replaced with data codes in the transcribed data, to ensure study site and participant confidentiality. Participants were allocated unique identifier codes (e.g., I001_C1 [interview 1, centre 1]). Electronic copies of the audio-recordings and interview transcripts were stored in a password-protected computer system on the secure university server. Only the researchers had access to the secure electronic files. 

### 2.9. Data Analysis

An inductive thematic analysis was undertaken [31], as follows. Data familiarisation was conducted independently by authors IH and LC, who read through the interview transcripts to obtain an overview of the depth, richness and diversity of the data collected. Transcribed interview data were inspected independently by IH and LC, who tagged participant responses (excerpts) that directly responded to the a priori interview questions. Data codes arising from these excerpts were independently developed by IH and LC and entered to data tables under each of the interview questions (see Appendix A). The relevant data codes and supporting excerpts were then tabulated under each research question and compared. Initial inter-rater reliability (IRR) of data codes independently allocated by IH and LC was calculated as the number of agreed codes over the total number of codes allocated, which achieved 82% agreement [32]. Author AW independently reviewed the selected transcript excerpts and the codes allocated by IH and LC. All authors then discussed and agreed on the naming and allocation of the codes. Using the agreed codes, IH, LC and AW independently performed two further iterations of coding analysis, achieving an IRR of 86% using the following formula: dividing the number of agreed codes by the number of agreements and disagreements [32]. The common themes arising from the agreed codes were identified and named by IH, LC and AW through close inspection and discussion of the participant excerpts associated with the agreed data codes [31]. The methods of the study adhered to the Standard for Reporting Qualitative Research [33] (see Appendix A).

## 3. Results

### 3.1. Participant Characteristics

Participants comprised 13 community-dwelling adults aged 69–94 years with at least one sensory change (see Appendix A). 

### 3.2. Sensory Changes Identified

Vision and hearing impairments were the most commonly identified sensory changes. Participants with visual changes reported inability to focus on near objects, having double vision and impaired night vision. Five participants reported past or current cataracts, three reported macular degeneration and two reported glaucoma. Participants with hearing impairment reported decreased hearing acuity, speech intelligibility, pitch discrimination and reduced hearing threshold. Four participants discussed changes in taste, two identified changes in smell and two identified changes in touch which was reportedly manifested by weakened grip strength and reduced stability when walking.

### 3.3. Interview Question 1: To What Extent Is Sensory Change Apparent to Older People?

#### 3.3.1. Apparent over Time

Nine participants advised that sensory changes became apparent over time, where it “was gradual” (I011_P) and “got progressively, slowly, worse” (I001_C1).

#### 3.3.2. Not Initially Apparent

Seven participants were initially unaware of the sensory changes occurring.

“…if you don’t hear something, you don’t hear it, you don’t know you don’t hear it.”(I001_C1)

These changes only became apparent when identified by family members/carers or doctors, or when realising that their behaviour had changed such as “talking a bit loud” (I006_C4).

“I only realised (hearing loss) when he (husband) told me because he was very soft-spoken.”(I002_C1)

#### 3.3.3. Became Apparent with a Particular Situation

Three participants could pinpoint a situation that made them suddenly aware of their sensory changes, for example a reduced ability to engage with others in social settings, or when they were unable to hear dialogue on television. 

“I was having a very intimate intense conversation, I suddenly realised later I wasn’t sure whether I was saying yes or no at the right time.”(I004_C3)

### 3.4. Interview Question 2: Which of the Senses Is the Most Important to Older People?

Eight participants indicated that eyesight was the most important sense, because eyesight is “one of the main things of life” (I010_C4) and it would be difficult to function without it. 

“I think I’d rather put up with anything else but my eyes last cause once you lose your eyesight you can’t do anything.”(I009_C4)

“It would be really difficult to get around without eyesight.”(I003_C2)

The remaining five participants responded according to the sense that had the most impact on their quality of life, and this was mainly hearing. 

“…deafness, deafness is in the family and uh I just have to live with that, if I didn’t have hearing aids I wouldn’t be able to hear you… it is both ears”(I004_C3)

“… when I calculate the impact, it is profound”(I001_C1)

For one person with multiple sensory changes, it was paraesthesia (touch) which had the greatest impact on quality of life.

“…the pain in my leg comes from the ankle up to here to my bottom. And it is like a knife, like a you know lighting strike and it goes there and oh my god, I think heavens it is really bad”(I006_C4)

### 3.5. Interview Question 3: What Is the Older Person’s Experience of Living with Sensory Change?

All 13 participants identified that some sensory changes impacted on important aspects of their life. In acknowledging the limitations imposed by sensory changes, many of them took measures to “try to make the most of it” (I004_C3). The most important consideration for all participants was to live a self-determined, independent life in the presence of sensory impairment.

#### 3.5.1. Impact on Independence in Activities of Living

All 13 participants had vision changes and discussed the negative impact of declining vision on daily functioning. Impacts included lack of ability and opportunity to continue driving, and loss of ability to do their own shopping without the assistance of others. Maintaining independence in daily living activities outside of the home was considered an important function of self-determination.

“I am happy living independently…because I can read and write. If I cannot do that then I become dependent. And my independence will go.”(I002_C1)

#### 3.5.2. Impacts on Established Lifestyle

For a small number of participants, sensory changes also reduced opportunities for ongoing employment due to inability to perform tasks satisfactorily and thus, hastened retirement. 

“I had to give up lecturing because to teach I ask questions, it’s interactive business when you are a teacher like I am. I just couldn’t hear what they were saying.”(I001_C1)

For 11 participants, vision and/or hearing changes reduced engagement in indoor leisure activities including listening to music, reading and watching television. These impacts challenged their self-identity, and some participants spoke about wanting to feel young again. 

“…nothing lasts forever, everyone says you look young, … I wish I could feel the feeling.”(I007_C4)

#### 3.5.3. Impacts on Social Life

Eight out of 11 participants with hearing changes chose to retreat into an inner world by avoiding social interactions, particularly large groups, as “it is chaos” (I001_C1). They found it difficult to discriminate conversation with one person in noisy social situations. This type of sensory overload caused some participants to prefer “living in silence” (I004_C3). 

“I avoid gatherings and that’s bad for someone…who is used to large groups.”(I001_C1)

“I don’t talk when I am in the van with the others because if they can’t hear me and I can’t hear them, then we will be making a mess, so it is best to be quiet.”(I008_C4)

#### 3.5.4. Impacts on Emotions and Emotional Response

Negative emotional (affective) responses to sensory changes were commonly reported by participants. Eight participants expressed irritation when unable to participate in social activities because of multiple and overwhelming sensory stimuli, such as several people speaking at the same time. Others expressed feelings of agitation “because they (family) do not understand [difficulties with joining in conversation]” (I001_C1) resulting in strained family relationships. Six participants felt resigned to the changes, helpless and discouraged when hoped-for improvements were not achieved with medical treatment. 

“Well I would like to hear better. I am not satisfied but I’ve tried to improve, and it didn’t, so what can I do.”(I003_C2)

Consequently, these participants felt “disabled” (I003_C2) by their continued sensory impairment. 

#### 3.5.5. Increased Dependence on Family Members

All participants reported becoming increasingly dependent on family members or friends in completing activities of daily living (e.g., for medication support and transport to specialist appointments or shopping). For the 11 participants who lived alone, emotional support was reported to be received by family via email or phone, to gain “a source of energy” (I004_C3). Some participants reported appreciating the practical support provided by families, friends and community outreach services, to help them with their shopping, to attend medical appointments and to get prescriptions filled. Whilst other participants reported to reluctantly receive assistance from family or friends. 

“My daughter-in-law gets them (medications) ready for me and I got to have them in the morning and at night before I go to bed.”(I010_C4)

#### 3.5.6. Taking Measures to Retain Self-Determination

Ten participants reportedly took measured risks to maintain their self-determination and independence, including living in potential hazardous environments (alone and up flights of stairs), not taking safety measures to accommodate their impairment (e.g., finding their way to the toilet in the dark), not paying attention to other sensory changes and choosing not to use sensory aids (mainly for hearing). Six out of 11 participants with hearing impairment chose not to use hearing aids because “it made little difference” (I009_C4), the aids were found to amplify background noises which they “can’t stand” (I007_C4) and some individuals found it difficult to physically manipulate the aids. These participants did not appear to acknowledge the risks associated with choosing not to use hearing aids inside and outside of the home situation, e.g., inability to hear fire alarms and traffic. 

### 3.6. Interview Question 4: How Do Older People Respond to and Manage Sensory Change?

While many of the participants tended to ignore the less impactful sensory changes (e.g., smell, taste and touch), when a significant change or loss of these senses prevented them from engaging in everyday living activities (e.g., food preparation, cooking, reading, watching television, going to the post-office), they did seek health professional assistance. All 13 participants consulted with health professionals and health services in response to significant visual impairment. They were more inclined to have regular check-ups for vision rather than hearing changes (varying between 2 and 12 months). Participants with hearing loss who used hearing aids visited their audiologists on an ad hoc basis (approximately once every 12 months), choosing to visit only when their hearing aid required adjustment. While some participants proactively sought medical review for vision and/or hearing issues, many participants reportedly relied on service recall for annual check-ups. 

#### 3.6.1. Adapting Positively to Sensory Changes

Six of the participants who appeared to have adapted positively to sensory changes, through goal-directed behaviour, focused their efforts on developing new social roles, such as volunteering work and participating in research on new treatments for sensory loss. 

“I volunteer at the National Acoustic Laboratories as a subject…. I’ve done that for about 6 to 8 years and through that I have learned a lot about acoustic engineering and so on and that is interesting…that is where a lot of my energy has been directed. Energies that used to go into full time job and teaching and so on.”(I001_C1)

Those participants that adapted positively to sensory changes also reported to make adjustments to continue to attend social functions and maintain social networks. 

“(Day centre) it’s taken my mind (off the sensory changes) … thinking different things, you meet other people and talk”(I007_C4)

“… you know, I’ve always had a listening career (as a psychotherapist and parish priest), I’ve always listened to people, I wanted to keep up my optimum…I learned to listen by other means as well. The whole sensory apparatus, yeah… you know, I can be deaf but listen…”(I004_C3)

Participants also “worked against being invalidated” (I004_C3) by a decline in their senses through championing the cause for improved sensory support, for example, by insisting that microphones were used in churches and that the hearing loop was switched on in lecture theatres. 

“…you have to make sure there is a loudspeaker running around. I had this church set up new microphones… we got it up…. I got a little device …for when I can turn on the T-Loop in church…(I004_C3)

Only two participants spoke directly about establishing goals in relation to sensory maintenance. One participant, who was particularly motivated to improve her hearing impairment, regularly participated in hearing research. The second participant considered how setting goals would help to improve sensory changes in relation to maintaining daily life activities. 

“…chances are I am thinking of what I need to, what I want to achieve the next day… I am awake I am thinking, thinking of what needs to be done. Then I am driven, there is this motivation”(I003_C2)

#### 3.6.2. Lacking Motivation to Plan and Act on Sensory Impairment

Seven of the 13 participants lacked the motivation to take action to improve their sense/s, despite reporting negative impacts on their life, as a result of the decline in sensory function. These participants did not set goals in relation to their sensory changes and reported a lack of motivation to improve their situation. In addition, often adaptation aids were unable to provide sufficient benefits to warrant sustained behaviour change. 

“…seems to be alright. You get lazy.”(I005_C4)

The high cost associated with treatment for sensory loss reduced some participants’ motivation to actively manage their sense/s.

“I was referred to a specialist, but I am having second thoughts because he is charging enormously for things that could be done for free.”(I003_C2)

When available sensory aids were not helpful, motivation to seek improvement waned.

“Do I set any goals? Well I should but...I don’t… I’ve been to two places to get hearing aids…. one I can’t use, and the other doesn’t make any difference…so no, I don’t”(I009_C4)

One participant gave voice to the sentiments expressed by those who lacked motivation to establish, plan and act on goals in managing their sensory change/s.

“…Oh no, no goals…I just try to do everything that I can and…. And you know…. there are more or less no goals…that’s it. But I live from day to day and that’s it, that’s all I can do now, can’t do anything more…(I006_C4)

## 4. Discussion

This qualitative descriptive study used semi-structured interviews with 13 volunteer community-dwelling adults 65 years and older to explore their perceptions related to the identification, lived experiences, responses and management of change/s in the senses of hearing, vision, touch, taste and smell. In agreement with the literature, sensory changes were often unnoticed by the person themselves, becoming apparent when having difficulties in specific social or work-related situations and/or when identified by others such as family, friends, work colleagues and doctors [3,34]. Most often, it was the comments of others about the obvious sensory loss that triggered attention to the deterioration that was occurring. It is commonly reported that age-related sensory change often occurs slowly, and the individual subsequently adjusts their behaviour to accommodate the declining abilities [2,10]. For example, participants in this study with poor vision had stopped reading and subsequently, no longer recognised the further deterioration in their vision in other areas of life.

Study participants were more aware of and placed greater emphasis on impaired vision and hearing, as compared to other senses, and decline in vision and hearing was given prominence and attention in both medical and self-management actions [2]. Participants considered that the dual loss of vision and hearing impacted most on their lifestyles, independence, well-being, quality of life and social interactions. Typically, changes in the somatosensory systems of taste and smell are not consistently identified by older persons compared to the more obvious hearing and vision changes [2]. Impairment of vision and hearing becomes more readily identifiable by older persons because they impact more profoundly on daily function, such as reading, shopping, driving and communication [2]. This was reflected by most of the study participants who identified vision to be the most important sense in terms of maintaining independence. Independence was also challenged by a loss of career, social life, function and self-esteem with dual vision and hearing impairment. 

Older adults with dual sensory impairment are also more likely to have other health issues, as compared to those with no impairment or uni-sensory impairment [35]. Dual hearing and vision impairment is a substantial marker for frailty in older age, thus a decline in either of these senses places older persons at elevated risk of numerous adverse health outcomes such as falls, institutionalisation, hospitalisation and disability [36,37]. Since dual vision and hearing loss is prevalent in the older-old population [35], these persons are at even greater risk of frailty and its sequelae [27]. Schneider et al. [38] estimated that 25 percent of participants aged 80 years and over in the Australian Blue Mountains Eye Study experienced combined vision and hearing loss. In that study, dual sensory impairment commonly affected how these older people lived and interacted with their physical and social environments. Our study concurs with those findings, as study participants advised that they withdrew from group communication-based situations and found themselves becoming socially isolated and reliant on family for social connections, shopping and medical appointments. Some of the participants articulated ways in which these sensory losses contributed to their feelings of social isolation and dependency. 

Only one participant independently and proactively mentioned other sensory changes related to taste, touch and smell, during the interview. When prompted about these sensory areas, a small number of participants identified experiencing changes in their sense of taste, which they equated with reduced eating pleasure and appetite, and altered dietary habits. Reduced eating pleasure and altered dietary habits are reported to be common motives for seeking medical attention [39]. However, Arganini and Sinesio [16] hypothesised that the role of chemosensory impairment (taste and smell impairment) in diminished eating pleasure and appetite is overestimated, and that psychosocial factors such as loneliness, dietary restrictions, perceived taste impairment and subjective health status significantly influence the decline of eating pleasure in older people. This hypothesis is also supported by other studies reporting the limited influence of taste and smell impairments on food liking or intake [40,41]. These findings indicate that psychosocial factors related to altered dietary habits should not be overlooked by health professionals. In addition, the findings of our study suggest that cultural factors may also influence recognition of altered taste. For example, two of the four study participants who reported a change in taste and identified the need to add strong spices to increase the flavour of food, were of an Indian background. Further research could explore the relationship between cultural background and responses to changes in taste.

An impaired sensation of touch, smell and taste can also impact on the older person’s function and their quality of life by influencing the way they experience the environment and react to stimuli [3]. For example, touch impairment increases the risk of falls and fractures [42], whereas olfactory impairment complicates the detection of dangers in the environment (e.g., smoke, gas, spoiled food) and may lead to changes in food choice [39]. A few participants discussed the impact that these sensory changes had on their lives, including experiencing a fall and sustaining subsequent injury, increasing salt in their diet despite a diagnosis of hypertension and regular occasions of not smelling burnt food while cooking. Given the impacts of touch, smell and taste on the individual’s functional independence and safety, it is concerning that most of the study participants were unaware about these sensory changes. Future studies may explore older people’s responses to changes in touch, taste and smell.

The question of how the participants responded to and managed sensory impairment/s sheds light on their reliance on health professionals and family/friends to provide reminders and support regarding sensory check-ups, use of sensory aids and strategies to accommodate sensory changes. Although most participants with visual impairment were reliant on different prescription glasses, used them as instructed, complied with regular visual check-ups and were relatively satisfied with the different glasses prescribed, the majority lacked the motivation to proactively plan regular check-ups and follow-up appointments. The participants who required follow-up support were reliant on reminders by optometrists and audiologists for ongoing assessment and treatment. A small number of these participants were reluctant, or refused, to seek specialist review which they considered too costly. These findings indicate the need for enhanced education and communication on the importance of follow up care and vision/hearing management with a health care provider for the older person who lacks motivation to maintain their senses. Family/carers need to be included in such conversations where older persons lack an appreciation of and a willingness to maintain sensory function [43]. Health services also need to streamline and support the attendance of the older person at follow up appointments. A variety of actions might facilitate this including services having a recall and reminder system, targeting and recalling patients who usually attend ad hoc for appointments, routine referral to an ophthalmologist and/or specialist audiologist for review and routine education of patients and their support persons on vision and hearing management and importance of reviews [20]. 

Taking a proactive stance is essential for older people with hearing impairment who consider further medical treatment or management is futile. Participants who reported having a hearing aid relayed difficulties in using the apparatus effectively and were generally dissatisfied with the ability of the aid to help with hearing, particularly in public situations. Since more sophisticated hearing technologies are becoming accessible to older people, it is important to understand why their uptake was so low in a majority of study participants with hearing loss [22]. The possible reasons for low uptake are the older person’s lack of knowledge and/or the value placed on perceived benefits of investing in such technology [26]. These factors were identified in the study, since participants were influenced by their awareness of particular hearing technology, their self-efficacy in using this technology and the facilitating conditions which were conducive to technology access. Thus, the general value of a particular technology and individual goals for improving their sensory impairment were insufficient to motivate participants to investigate and adopt advanced hearing technology.

Indeed, most of the participants lacked any thought of goal-directed behaviour to redress their hearing loss and were inconsistent in using their hearing aids, owing to their perceived ability to overcome contextual (environmental) barriers. Traditional hearing aid technology has not been found to meet consumer expectations and was generally considered inadequate for use in some situations [18,20]. As participants reliant on hearing aids advised, sound amplification can be an effective tool for enhancing the communication abilities of an individual in private conversation and quiet settings, but in group situations amplification is insufficient in managing sound discrimination [21]. Simply amplifying sound does not fully correct the auditory deficit when the defect is considered multifactorial in nature. As well, amplification can be distressing in noisy situations [44]. Furthermore, the high cost associated with specialist appointments and sensory aids was a barrier for participants to actively manage their sensory impairment. For the reasons identified, 7 of the 13 participants lacked motivation and did not set goals to improve hearing ability. 

In light of recent research on older people’s motivation to adopt self-service health technologies, this finding indicates that interventions to support hearing impairment also need to consider the views of the older person on what may be effective management approaches, and what value such technology makes to their everyday life [26]. Research needs to focus on more than improved features of hearing technology. Addressing the contextual constraints to the uptake of new technologies by older people might include psychosocial counselling for emotional and social adaptation, as well as environmental accommodations (e.g., installing and switching on hearing loops in institutions) and where relevant, rehabilitative hearing training to supplement hearing aid use [20]. As well, effort needs to be placed on convincing older people of the expected benefits and usefulness of new technologies by targeting their control beliefs (flexibility and economic benefits) and self-efficacy in its use [24,26]. These efforts might assist in motivating people with hearing loss to take a proactive stance in managing the impairment. 

According to the Goal-Directed Behaviour model [23], people experiencing health issues will demonstrate positive affect and adopt adaptive behaviours when the health demands correspond with the individual’s ability and motivation to address these demands. This occurred for 6 of the 13 participants, whose positive approach to sensory change was influenced by the process of goal-directed behaviour, which motivated them to devote time and energy into planning and taking action on sensory impairment/s impacting on their quality of life. On the other hand, for seven of the participants, both internal and external factors overwhelmed their sense of competence to deal with the sensory loss, resulting in apathy, despondency and lack motivation to seek a solution to the issue. The participants who lacked motivation to remedy or improve sensory change/s felt overwhelmed, or powerless, to address environmental and social constraints. Therefore, for these study participants, a significant moderator of the intention–behaviour relationship was the extent to which their motivation and intentions were based on anticipated affective reactions when seeking and finding further support for sensory change/s [34]. This was reflected in study participant responses when hoped-for improvements in sensory function were not achieved with further health review and management. Many of these participants were left feeling resigned, helpless and discouraged about these negative outcomes. Thus, the relationship between value perceptions and technology adoption requires more research to integrate the concepts of context, self-efficacy and utility in older people with sensory impairment.

### 4.1. Strengths and Limitations

A study strength included the participation of all three researchers in planning, designing and undertaking key components of the study. This collaboration was particularly fruitful when analysing and interpreting interview responses, following an agreed and consistent approach. This process ensured that investigator triangulation was achieved in data analysis, whereby the data were independently analysed by each of the researchers and then comprehensively interrogated before coming to a consensus on data codes and themes [45]. Reflexology was, therefore, achieved by attending systematically to the context of knowledge construction, especially to the effect of the researchers, at every step of the research process [46]. Additionally, the authors’ personal and professional experiences with older adult healthcare and gerontological knowledge were subject to considerable reflection when devising the research questions, making decisions on how these questions might be posed and probed at interview, and when reflecting on participant responses [32]. 

While the study was not framed specifically by a theoretical model, we were interested to discover whether goal-directed behaviour was relevant to the participants’ responses to sensory impairment and hence, their targeted management of the changes occurring. Some of the interview questions were, therefore, directed towards goal-directed behaviour in managing sensory impairment. The responses to these questions are illuminating and have provided a reference lens for future investigation of older persons’ management of health issues. Consequently, the findings of the current study have application beyond its context, such as seeking to understand and address older person’s variable approaches to chronic illness management [47]. In particular, the study’s novel data on the limitations of common management approaches to sensory impairment, especially with hearing loss, provide valuable information for improved service offerings and essential follow-up mechanisms. The participants’ knowledge, experiences and approaches to living with sensory impairment have resulted in an information-rich sample which correlates with the research questions posed [48].

One study limitation is that the participants’ cognitive and sensory status were not objectively assessed prior to interviews. Cognitive impairment was observed in some participants (e.g., going off tangent in interviews), which may have impacted on their ability to accurately respond to questions. Further studies could explore the relevance of cognitive impairment on experience of sensory change. Due to the participants’ age and fatigue level, some interview questions could not be probed to a great extent. For example, participants chose to focus on and discuss in more detail hearing and vision impairment. Consequently, questions regarding the sense of taste, touch and smell became a reduced focus in the interviews. Future research could benefit from conducting short interviews in multiple stages (different times or days), which would allow an equal focus on all senses during the interview process. Further exploration is warranted on factors affecting motivation of sensory maintenance, and the effectiveness of goal setting in this regard. 

### 4.2. Implications

Health professionals play an important role in assisting older people to manage sensory impairments. Correction of sensory deficits, where possible, is an important quality of life and safety issue in preventing frailty, falls and subsequent injury, and associated morbidities and mortality. The study findings indicate that an important health professional function is to routinely discuss the functioning of each of the five senses with all persons 65 years and older. Regular screening will facilitate early intervention and support. This might include asking the person what they are able and not unable to do in attending to activities of daily living, and what they have stopped doing, since older people may understate or find it difficult to fully describe the extent of their sensory impairment/s. The answers to these questions will give a clearer indication of the extent of deterioration of the sensory modality, and the associated impact on independence, physical and psychosocial function and life quality. Consequently, health professionals also need to identify how older people with sensory loss/es interact with others, seek details on their thoughts and feelings about the quality of those interactions and relationships, and the affect that these have on the older person’s physical and mental health and well-being.

Since motivation changes with age-related sensory impairment, it is important that health professionals recognise the importance of increasing sensory stimulus to increase the older person’s motivation in maintaining their physical and psychosocial function. While sensory changes need to be assessed by history taking, physical examination and clinical testing, psychosocial function and motivation need to be assessed by asking questions on the thoughts and feelings related to the sensory impairment. This can be determined by evaluating the presence and adequacy of family and social support networks, and by building interventions to improve the person’s motivation to improve or retain sensory function, based on their personality, capabilities and addressing impediments to motivations such as costs of services and sensory aids. Family members/carers can be encouraged to reinforce use of corrective aids and desired safety behaviour to older people. Finally, helping the person to develop realistic, clear and specific goals for maintaining their sense/s may help with motivation and forming a belief that the effort is worthwhile in retaining independence and achieving quality of life.

## 5. Conclusions

The presence of two or more sensory deficits constituted an important facet of the ageing process for the study participants. The findings suggest that in older age, the relationship between sensory impairments, health, function, independence and well-being is profound. In circumstances where the participants experienced ongoing sensory decline within the context of perceived futility of corrective treatment or aids, they easily became discouraged and withdrew from many aspects of social and family life. In most cases, study participants were not proactive in maintaining their senses, and this related to lacking the motivation to establish goals for planning and acting on sensory changes. Motivation to deal proactively with continuing sensory decline was partly a factor of long-range outcome expectations, possibly also their personal characteristics such as mood and personality, and their access to family, social and health service support, including costs of services and sensory aids. These factors need to be considered by health professionals when supporting older people to maintain sensory function.

## Data Availability

The data presented in this study are available in Appendix A.

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
