# Peer review of "Older People’s Experiences of Living with, Responding to and Managing Sensory Loss"

_healthcare, 2021, doi:10.3390/healthcare9030329_

Round 1

Reviewer 1 Report

Thank you for the opportunity to review this interesting paper on a qualitative study  through face-to-face semi-structured interviews with a convenience sample of thirteen community-dwelling adults 65 years and older. I would like to congratulate authroa as I found very well structed the manuscript.

I only have minor points that I hope it could be of help to improve the current manuscript. I would extend discussion on the applied level, eg., telehealth medicine or technology adoption for this profiles. E.g.,

These results might reflect reasons of underuse or even inappropriate application of devices for telemedicine or even training proposes in technology adoption, as indicated by:

Czaja, S. J., Charness, N., Fisk, A. D., Hertzog, C., Nair, S. N., Rogers, W. A., & Sharit, J. (2006). Factors predicting the use of technology: findings from the Center for Research and Education on Aging and Technology Enhancement (CREATE). Psychology and aging21(2), 333.

Moret-Tatay, C., Beneyto-Arrojo, M. J., Gutierrez, E., Boot, W. R., & Charness, N. (2019). A spanish adaptation of the computer and mobile device proficiency questionnaires (CPQ and MDPQ) for older adults. Frontiers in psychology10, 1165.

McGrath, C., & Corrado, A. M. (2019). The environmental factors that influence technology adoption for older adults with age-related vision loss. British Journal of Occupational Therapy82(8), 493-501.

Author Response

Dear Reviewer,

Thank you very much for taking the time to review our manuscript. Please find the response to your comments below:

Point 1: I only have minor points that I hope it could be of help to improve the current manuscript. I would extend discussion on the applied level, eg., telehealth medicine or technology adoption for this profiles. These results might reflect reasons of underuse or even inappropriate application of devices for telemedicine or even training proposes in technology adoption. 

Response 1: Thank you for your suggestion. We have updated the manuscript to consider motivation and the use of self-service technology for sensory impairment in older people. The changes can be found in the introduction section (page 2, lines 85-102) and discussion section (pages 11-12, lines 528-566).

Reviewer 2 Report

I commend the authors on producing an insightful manuscript. It is well presented and results / discussion provide good depth of analysis. The method is clear and gives a comprehensive overview of the study. The authors provide a good amount of rich description within the results, allowing for the meaning of the participants own words to be maintained. There are just two main points that I would like to see included:

  1. Could the authors clearly articulate how trustworthiness was established in the qualitative procedure.
  2. The inclusion of a visual / figure detailing the raw data and higher order themes that were identified would add to the results

Author Response

Dear Reviewer,

Thank you very much for taking the time to review our manuscript. Please find the response to your comments below:

Point 1: Could the authors clearly articulate how trustworthiness was established in the qualitative procedure

Response 1: Thank you for your comment. The methods of this study adhered to the Standards for Reporting Qualitative Research (SRQR). We have added this information in the data analysis section (page 5, lines 244-245). We have uploaded the SRQR checklist in Supplementary material 3. In the SRQR checklist, there is a section on “techniques to enhance trustworthiness”. We have noted down the pages and lines which correspond to this section in the checklist (Page 5, Lines 225-245 and Page 12, Lines 588-600). We have also added that investigator triangulation was achieved in the data analysis process (page 12, line 591).

Point 2: The inclusion of a visual / figure detailing the raw data and higher order themes that were identified would add to the results

Response 2: Thank you for your comment. As the data code and thematic tables are quite detailed for each research question, we have uploaded it as Supplementary material 2.